# CONFORMAL STRUCTURED PREDICTION

**Botong Zhang, Shuo Li & Osbert Bastani**
Computer and Information Science
University of Pennsylvania
{bzhang16, lishuo1, obastani}@seas.upenn.edu

## ABSTRACT

Conformal prediction has recently emerged as a promising strategy for quantifying the uncertainty of a predictive model; these algorithms modify the model to output sets of labels that are guaranteed to contain the true label with high probability. However, existing conformal prediction algorithms have largely targeted classification and regression settings, where the structure of the prediction set has a simple form as a level set of the scoring function. However, for complex structured outputs such as text generation, these prediction sets might include a large number of labels and therefore be hard for users to interpret. In this paper, we propose a general framework for conformal prediction in the structured prediction setting, that modifies existing conformal prediction algorithms to output structured prediction sets that implicitly represent sets of labels. In addition, we demonstrate how our approach can be applied in domains where the prediction sets can be represented as a set of nodes in a directed acyclic graph; for instance, for hierarchical labels such as image classification, a prediction set might be a small subset of coarse labels implicitly representing the prediction set of all their more fine-descendants. We demonstrate how our algorithm can be used to construct prediction sets that satisfy a desired coverage guarantee in several domains.

## 1 INTRODUCTION

Deep neural networks (DNNs) have recently proven to be highly effective at solving challenging prediction problems. Despite this progress, a key challenge is the difficulty building reliable systems out of DNNs since they are intrinsically prone to error. One way to address this challenge is to use uncertainty quantification to determine when the model's predictions may be unreliable. As a consequence, uncertainty quantification has been a key strategy for improving reliability when integrating DNNs into broader systems or when interfacing with human experts.

Conformal prediction has emerged as a promising strategy for uncertainty quantification (Vovk et al., 2005; Angelopoulos et al., 2023). This technique replaces a model $f : \mathcal{X} \to \mathcal{Y}$ with a *conformal predictor* $h : \mathcal{X} \to 2^{\mathcal{Y}}$; given an input $x \in \mathcal{X}$, $h(x) \subseteq \mathcal{Y}$ is a set of labels that captures the uncertainty of the model. One of the key advantages of conformal prediction is that it provides a *coverage guarantee*—roughly speaking, $h(x)$ is guaranteed to contain the ground truth label $y^*$ corresponding to $x$ with high probability under standard assumptions. This strategy provides the user with an interpretable form of uncertainty quantification. For instance, suppose a robot is using an object detector to identify obstacles; if we apply conformal prediction to the object detector, then the robot could avoid the entire prediction set of obstacles to ensure safety with high probability.

Most existing conformal prediction algorithms target the classification and regression settings, where prediction sets have simple structures—e.g., a subset of labels in classification, or prediction intervals in regression. However, many practical prediction problems involve far more complex structured outputs. While we can naïvely apply existing algorithms to these problems, the resulting prediction sets ignore the structure of the label space, which can lead to prediction sets that are large and uninterpretable. As a consequence, there has been recent interest in developing conformal prediction algorithms that are tailored to structured prediction; however, existing approaches have all targeted specific domains, such as code generation (Khakhar et al., 2023) and question answering (Mohri & Hashimoto, 2024; Quach et al., 2024), and do not provide general algorithms.

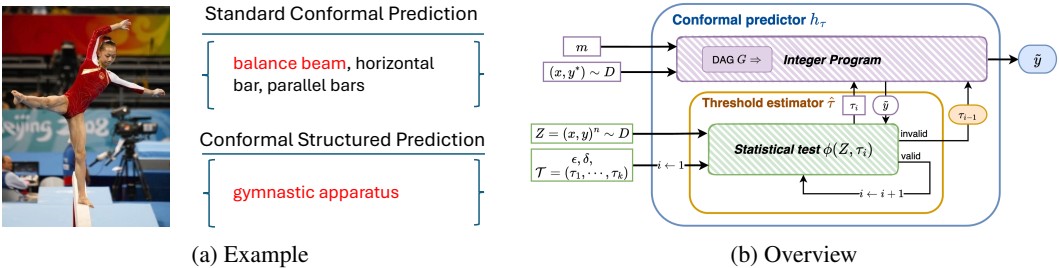

(a) Example            (b) Overview

Figure 1: (a) Structured prediction sets improve interpretability while maintaining the coverage guarantee. In this example, the standard conformal prediction set (top) is guaranteed to include the true label "balance beam" with high probability, but may be more difficult to interpret for someone without gymnastics knowledge. In contrast, the structured prediction set (bottom) can be more interpretable since it contains only a single coarse-grained label "gymnastic apparatus", while guaranteeing that the true label is descendant of this label in the label hierarchy with high probability. The error level for both the standard conformal prediction and conformal structured prediction is $0.05$ (i.e., the desired coverage level is $0.95$). See Section 5.2 and Appendix A.3 for more examples. (b) An overview of our framework. To estimate the conformal predictor parameter $\tau$, our algorithm uses a statistical test $\phi$ designed to either establish marginal or PAC coverage guarantees based on the given calibration set. It iterates until an invalid $\tau_i$ is identified, and returns the last valid threshold $\tau_{i-1}$. This computation assumes given a subroutine to compute the optimal prediction set $\tilde{y}$. In general, any optimizer can be used in conjunction with our framework; for the case where the prediction sets are derived from a DAG structure (including hierarchical labels), we show how the optimization problem can be encoded as an integer program.

We propose a novel framework for conformal structured prediction. Our goal is to generate *structured prediction sets*, which are interpretable representations of a potentially large number of concrete labels. For instance, in image classification, a structured prediction set might be a small set of coarse-grained labels that implicitly represent the set of all fine-grained labels that are leaf nodes of the coarse-grained labels in the label hierarchy (see Figure 1a). Alternatively, for a question answering task where the answer is a year, a structured prediction set might be a small set of intervals.

As with traditional conformal prediction, our framework considers a search space over conformal predictors parameterized by a single real number $\tau \in \mathbb{R}$, which intuitively says that the prediction set should include labels with predicted probability at least $\tau$. However, whereas in traditional conformal prediction, the search space over $\tau$ is monotone (i.e., coverage always decreases in $\tau$ whereas prediction set size always decreases in $\tau$), this is no longer the case in the structured prediction set setting. Existing conformal prediction algorithms make significant use of this monotonic structure to enable efficient estimation of the optimal choice of $\tau$. Our framework modifies these existing algorithms using an approach inspired by techniques from the learn-then-test framework (Angelopoulos et al., 2022) (which is designed to handle potential lack of monotonicity due to non-standard choice of loss function). Roughly speaking, our algorithm searches sequentially over the space of candidate $\tau$ and returns as soon as it finds one that is invalid. For achieving marginal coverage guarantees, the correctness guarantee follows directly from learn-then-test. We extend these ideas to the setting of probably approximately correct (PAC) (or training conditional) coverage guarantees (Vovk, 2012; Park et al., 2020). In particular, we provide a strategy that can be applied to obtaining PAC prediction sets in the structured prediction setting.

Next, we instantiate our framework in the context of a general class of structured prediction sets. We consider structured prediction sets that are subsets of a directed acyclic graph (DAG). For instance, the DAG might represent the label hierarchy, where the internal nodes are coarse-grained labels and the leaf nodes are fine-grained labels. Then, a structured prediction set might be a small subset of coarse-grained labels, which corresponds to the prediction set of fine-grained labels that are descendants of those coarse-grained labels. In this setting, we describe how to compute the optimal structured prediction set for a given value of $\tau$ by expressing the optimization problem as an integer program. This algorithm can then be used both in conjunction with our framework to estimate the optimal parameter $\tau$, as well as during inference to compute structured prediction sets.

We empirically evaluate our approach in five domains: (i) predicting integers represented by a list of MNIST digits (LeCun & Cortes, 2010), where the structured prediction sets are ranges, (ii) hierarchical image classification using the ImageNet dataset (Deng et al., 2009), (iii) a question answering benchmark based on the SQuAD dataset (Rajpurkar et al., 2016) restricted to questions where the answers are years, and the structured prediction sets are unions of a small number of intervals, (iv) a Python code generation task based on the MBPP dataset (Austin et al., 2021), where the structured prediction sets consist of partial programs (i.e., programs with underline{holes} indicating portions that remain to be implemented), and (v) predicting emotion labels in a given piece of text based on the GoEmotions dataset (Demszky et al., 2020). Our experiments demonstrate how our approach can be used to construct small prediction sets while satisfying a desired coverage guarantee (marginal or PAC).

**Contributions.** We propose the first general framework (summarized in Figure 1b) for conformal prediction applied to structured label spaces. We assume the user provides a search space of interpretable structures that implicitly represent prediction sets of labels. Then, our framework constructs a conformal predictor that is guaranteed to achieve the desired coverage guarantee (either marginal or PAC) while attempting to minimize prediction set size.

**Related work.** Traditional conformal prediction algorithms provide a *marginal coverage guarantee* (Vovk et al., 2005; Angelopoulos et al., 2023), which says that assuming examples are drawn i.i.d. from the data distribution, then the prediction set contains the true label with high probability:

$$\mathbb{P}_{Z \sim D^n, (x,y^*) \sim D}[y^* \in h_Z(x)] \geq 1 - \epsilon,$$

where $x$ is the input, $y^*$ is the true label, $D$ is the data distribution, $Z$ is the calibration set provided to the conformal prediction algorithm, $h_Z$ is the conformal predictor constructed using $Z$, and $\epsilon \in \mathbb{R}_{>0}$ is a given error bound. Alternate algorithms disentangle the probability over $Z$ from the one over $(x, y^*)$ to provide *probably approximately correct (PAC)* (or *training conditional*) coverage:

$$\mathbb{P}_{Z \sim D^n} \left[ \mathbb{P}_{(x,y^*) \sim D}[y^* \in h_Z(x)] \geq 1 - \epsilon \right] \geq 1 - \delta.$$

Conformal prediction has recently been applied to deep learning, including image classification (Angelopoulos et al., 2023; Park et al., 2020), anomaly detection (Li et al., 2022a), object detection (Li et al., 2022b; Andéol et al., 2023), semantic segmentation (Angelopoulos et al., 2022), and question answering (Li et al., 2024). However, in these domains, the label space is either a finite set (i.e., classification) or a real number (i.e., regression), or a relatively simple product of the two. For instance, in object detection, the label is a list of tuples $(x_1, y_1, x_2, y_2, c)$, where $(x_1, y_1, x_2, y_2) \in \mathbb{R}^4$ are real-valued coordinates and $c \in \mathcal{C}$ is an object category. We can straightforwardly quantify uncertainty separately for each component using traditional conformal prediction.

However, in many domains, we may require alternative representations of the prediction sets to ensure interpretability. For instance, a natural way to represent uncertainty for hierarchical label spaces is to use coarse-grained labels (see Figure 1a for an example). Mortier et al. (2022) takes this approach, but unlike our method, they do not offer formal coverage guarantees. Furthermore, their goal is to find the prediction set with the highest probability mass, whereas our goal is to find the smallest possible prediction sets under a constraint that the coverage rate meets a desired level. Therefore, their approach sometimes obtains significantly lower coverage (e.g., $\leq 50\%$), whereas our results demonstrate that our approach achieves the desired coverage rate. Khakhar et al. (2023) provides a conformal prediction algorithm where the true label is a program, and the prediction set is implicitly represented as a "partial program" where certain portions have been omitted. Quach et al. (2024) and Mohri & Hashimoto (2024) adopt similar strategies for certain question answering tasks. However, these existing approaches are highly specialized to a specific form of prediction set, whereas our approach is general. Moreover, Quach et al. (2024) still use sets of samples, which may suffer from lack of interpretability. Mohri & Hashimoto (2024)'s algorithm is closely related to the learn-then-test algorithm. However, they do not consider a tree structure of structured prediction sets; instead, they only consider a single sequence of increasingly coarse-grained labels. Thus, their approach is not applicable to our problem.

Next, Angelopoulos et al. (2024) considers hierarchical image classification in ImageNet. However, their goal is to minimize more general risk functions instead of prediction set size, whereas our goal is to extend conformal prediction to structured prediction. In particular, their purpose is to control an alternative risk function based on the hierarchical label structure, while still constructing traditional prediction sets. Indeed, we believe that our approach could be combined with theirs to further improve interpretability of the resulting prediction sets. For this specific task, their algorithm

can be viewed as producing a structured prediction set; however, even in this context, their search is constrained to only parent nodes of the class with the highest estimated probability $\hat{y}$, whereas our search space is much more general (and can be much more flexibly specified by the user). Angelopoulos et al. (2021) proposes an algorithm that introduces regularization to encourage smaller and stable sets. However, the goal of their paper is to reduce the average prediction set size by adding a regularization term, whereas our goal is to compute structured prediction sets.

More broadly, conformal prediction has been worked with other kinds of structure. For instance, Lee et al. (2024) and Dunn et al. (2022) address cases where inputs are grouped and where repeated measurements are present in the dataset, respectively. However, these are examples of structure in the input space instead of the label space. Finally, Angelopoulos et al. (2022) studies minimizing risk functions other than prediction set size; though they are studying a very different problem, their techniques can also be applied to our problem when aiming to achieve marginal guarantees. To the best of our knowledge, we are the first to adapt these techniques to provide PAC guarantees.

## 2 PROBLEM FORMULATION

As with existing conformal prediction algorithms, we assume that the model $f : \mathcal{X} \to \mathcal{Y}$ is implicitly represented by a *scoring function* $g : \mathcal{X} \times \mathcal{Y} \to \mathbb{R}$, so $f(x) = \arg\max_{y \in \mathcal{Y}} g(x, y)$. Typically, $g(x, y)$ is the predicted probability that the true label for input $x$ is $y$, but we make no assumptions about $g$.

Given $g$, we consider a space of *structured prediction sets* $\tilde{\mathcal{Y}}$, along with a mapping $\gamma : \tilde{\mathcal{Y}} \to 2^{\mathcal{Y}}$ such that $\gamma(\tilde{y}) \subseteq \mathcal{Y}$ is a prediction set of labels, and a *size function* $\sigma : \tilde{\mathcal{Y}} \to \mathbb{R}$ measuring the size of a structured prediction set. We consider conformal predictors $h_\tau : \mathcal{X} \to \tilde{\mathcal{Y}}$ of the form

$$h_\tau(x) = \underset{\tilde{y} \in \tilde{\mathcal{Y}}}{\arg\min}\, \sigma(\tilde{y}) \text{ subj. to } \sum_{y \in \mathcal{Y}} g(x, y) \cdot \mathbb{1}(y \in \gamma(\tilde{y})) \geq \tau, \tag{1}$$

where $\tau \in \mathbb{R}$ is a real-valued parameter that we need to estimate. In other words, we want the smallest prediction set according to the size function $\sigma$ that achieves cumulative score at least $\tau$. Intuitively, if $g(x, y)$ is the predicted probability of $y$ given $x$, then the constraint says that $\tilde{y}$ covers $\tau$ fraction of labels $y \in \mathcal{Y}$ weighted by their probability $g(x, y)$. In general, while $g$ does not need to be a predicted probability, our approach may be more effective when this is the case.

The main challenge is estimating the parameter $\tau$. We assume given a held-out *calibration set* $Z \subseteq (\mathcal{X} \times \mathcal{Y})^n$ of i.i.d. samples $(x, y^*) \sim D$ from the underlying data distribution $D$. Then, we want to choose the smallest possible $\tau$ subject to some kind of coverage guarantee; i.e., $y^* \in h_\tau(x)$ with high probability assuming $(x, y^*) \sim D$. Specifically, we consider two coverage guarantees. First, we consider a *marginal guarantee*

$$\mathbb{P}_{(x,y^*) \sim D, Z \sim D^n}[y^* \in \gamma(h_{\hat{\tau}(Z)}(x))] \geq 1 - \epsilon,$$

where $\hat{\tau} : (\mathcal{X} \times \mathcal{Y})^* \to \mathbb{R}$ is an estimator outputting a choice of $\tau$ for a given $Z \in (\mathcal{X} \times \mathcal{Y})^*$, and $\epsilon \in \mathbb{R}_{>0}$ is a given error bound (note that $\tau$ depends implicitly on $\epsilon$). Second, we consider a *probably approximate correct (PAC) guarantee* (or *training-conditional guarantee*)

$$\mathbb{P}_{Z \sim D^n}[\mathbb{P}_{(x,y^*) \sim D}[y^* \in \gamma(h_{\hat{\tau}(Z)}(x))] \geq 1 - \epsilon] \geq 1 - \delta,$$

where $\epsilon, \delta \in \mathbb{R}_{>0}$ are given error bounds. Intuitively, PAC guarantees "disentangle" the probability over the calibration set $Z$ and the current example $(x, y^*)$; they are useful when the goal is to provide a $1 - \epsilon$ guarantee across all predictions with high probability.

Finally, given $\tau$, computing $h_\tau$ is nontrivial in general. We can typically formulate the computation of $h_\tau$ as a constraint solving problem such as an integer program; we describe how to do so for a special case in Section 3. Our estimators $\hat{\tau}$ require that $h_\tau$ is computed using the same algorithm on the calibration examples as on the new examples, but do not make any other assumptions; for instance, it could rely on heuristics instead of finding the global optimum.

## 3 ALGORITHMS FOR STRUCTURED CONFORMAL PREDICTION

Our algorithm considers a finite set of candidate thresholds $\mathcal{T}_\epsilon^* \subseteq \mathbb{R}$. We assume that $\mathcal{T} = (\tau_1, \tau_2, ..., \tau_k)$ is ordered by $\tau_1 > \tau_2 > ... > \tau_k$. We assume these thresholds are in descending

order (i.e., from least to most desirable). At a high level, our algorithm will consider each candidate threshold $\tau_i$ in sequence, using a statistical test in conjunction with the calibration set $Z$ to determine if $\tau_i$ is valid. We use $\phi : (\mathcal{X} \times \mathcal{Y})^* \times \mathbb{R} \to \{0, 1\}$ to denote this statistical test; its first argument is the calibration set $Z$, its second argument is a candidate threshold $\tau$, and its output is $\phi(Z, \tau) = 1$ if it determines $\tau$ is valid and $\phi(Z, \tau) = 0$ otherwise. We provide tests for marginal and PAC coverage.

Now, our algorithm considers increasingly desirable candidate thresholds $\tau_i$; for each one, it runs the statistical test $\phi(Z, \tau_i)$ to determine whether $\tau_i$ is valid. The search halts when $\phi(Z, \tau_i) = 0$ for the first time at $\tau_i$, and the algorithm returns $\tau_{i-1}$. Formally, given an error level $\epsilon \in [0, 1]$ and $m \in \mathbb{N}$, our algorithm returns the threshold $\hat{\tau}(Z) = \tau_{\hat{i}(Z,\phi)}$, where

$$\hat{i}(Z, \phi) = \arg\max_{i \in [k]} i \text{ subj. to } \bigwedge_{i'=1}^{i} \mathbb{1}(\phi(Z, \tau_{i'}) = 1).$$

In other words, $\hat{i}$ is the largest $i \in [k]$ before which all $\tau_{i'}$'s are valid. Below, we describe specific implementations of the statistical test $\phi$ for marginal and PAC guarantees.

**Marginal guarantees.** Given $\epsilon \in [0, 1]$, we use the following statistical test:

$$\phi_{\text{marginal}}^{\epsilon}(Z, \tau) = \mathbb{1}\left( \sum_{(x,y^*) \in Z} \mathbb{1}(y^* \notin \gamma(h_\tau(x))) \leq (n+1)\epsilon \right).$$

We have the following marginal coverage guarantee:

**Theorem 3.1.** *The estimator* $\hat{\tau}(Z, \epsilon) = \tau_{\hat{i}(Z, \phi_{\text{marginal}}^{\epsilon})}$ *satisfies*

$$\mathbb{P}_{Z \sim D^n, (x,y^*) \sim D}[y^* \in \gamma(h_{\hat{\tau}(Z,\epsilon)}(x))] \geq 1 - \epsilon.$$

*Proof.* This result follows from the learn-then-test algorithm (Angelopoulos et al., 2022). □

**PAC guarantees.** Given $\epsilon, \delta \in [0, 1]$, consider the statistical test

$$\phi_{\text{PAC}}^{\epsilon,\delta}(Z, \tau) = \mathbb{1}\left( \sum_{(x,y^*) \in Z} \mathbb{1}(y^* \notin \gamma(h_\tau(x))) \leq \hat{\ell} \right),$$

where

$$\hat{\ell} = \arg\max_{\ell \in \mathbb{N}} h \text{ subj. to } F(\ell; n, \epsilon) < \delta,$$

where $F(\ell; n, p) = \sum_{j=0}^{\ell} \binom{n}{j} p^j (1-p)^{n-j} < \delta$ is the cumulative distribution function (CDF) of the random variable $\text{Binomial}(n, p)$. Then, we have the following guarantee:

**Theorem 3.2.** *The estimator* $\hat{\tau}(Z, \epsilon, \delta) = \tau_{\hat{i}(Z, \phi_{\text{PAC}}^{\epsilon,\delta})}$ *satisfies*

$$\mathbb{P}_{Z \sim D^n}[\mathbb{P}_{(x,y^*) \sim D}[y^* \in \gamma(h_{\hat{\tau}(Z,\epsilon,\delta)}(x))] \geq 1 - \epsilon] \geq 1 - \delta.$$

*Proof.* Let $i_0$ be the smallest index of $\tau_i$ such that $\tau_i$ is invalid (i.e., $\tau \notin \mathcal{T}_\epsilon^*$), and let $\tau_0 = \tau_{i_0}$. Also, let $z = \mathbb{1}(y^* \notin h_{\tau_0}(x))$; note that $z$ is a function of the random variable $(x, y^*) \sim D$, and in particular $z \sim \text{Bernoulli}(\mu)$ with $\mu = \mathbb{P}[y^* \notin h_{\tau_0}(x)]$. Since $\tau_0$ is invalid, we have $\mu > \epsilon$.

Next, the sum in $\phi_{\text{PAC}}^{\epsilon,\delta}$ is a sum of i.i.d. samples from $\text{Bernoulli}(\mu)$. In particular, let $Z = \{(x_i, y_i^*)\}_{i=1}^n$, let $z_i = \mathbb{1}(y_i^* \notin \gamma(h_{\tau_0}(x_0)))$, and let $b = \sum_{i=1}^n z_i$; note that $b$ is a sum of $n$ i.i.d. Bernoulli random variables with mean $\mu$, so $b \sim \text{Binomial}(n, \mu)$. Now, $\phi_{\text{PAC}}^{\epsilon,\delta}$ has form

$$\mathbb{1}(\phi_{\text{PAC}}^{\epsilon,\delta}(Z, \tau) = 1) = \mathbb{1}(b \leq \hat{\ell}),$$

so

$$\mathbb{P}(\phi_{\text{PAC}}^{\epsilon,\delta}(Z, \tau) = 1) = \mathbb{P}(b \leq \hat{\ell}) = F(\hat{\ell}; n, \mu) \leq F(\hat{\ell}; n, \epsilon) < \delta,$$

where the first inequaltiy follows since the CDF of the $\text{Binomial}(n, p)$ is monotonically decreasing in $p$ and $\mu \leq \epsilon$ (this follows since $\frac{\partial}{\partial p} F(\ell; n, p) \leq 0$).

Finally, note that our algorithm returns a valid parameter $\tau_i$ as long as $\phi_{\text{PAC}}^{\epsilon,\delta}(Z, \tau) = 0$, since it returns $\tau_i$ for some $i < i_0$, and by definition of $i_0$, all of these $\tau_i$ are valid. Thus, our algorithm returns a valid $\tau_i$ with probability at least $1 - \delta$, as claimed. □

## 4 APPLICATION TO DAG STRUCTURED PREDICTION SETS

**Problem formulation.** We consider a prediction problem where the prediction set consists of selecting a subset of nodes in a directed acyclic graph (DAG). For example, the DAG may be a tree representing hierarchical labels. Consider a DAG $G = (V, E)$ with vertices $V = [k] = \{1, ..., k\}$ and directed edges $E \subseteq V \times V$. For example, for image classification, $G$ might encode a hierarchy of image categories. We let $L \subseteq V$ denote the leaf nodes of $G$; we assume that each $v \in L$ is labeled with a probability $p_v$, such that $\sum_{v \in L} p_v = 1$. In our image classification example, a leaf $v$ corresponds to a fine-grained label, and $p_v$ might be the predicted probability of that label.

Now, we consider structured prediction sets of the form $\tilde{y} \subseteq V$, where $|\tilde{y}| \leq m$ for a given hyperparameter $m$. Furthermore, we assume that the size function is $\sigma(\tilde{y}) = |\text{leaves}(\tilde{y})|$, where

$$\text{leaves}(\tilde{y}) = \{v \in L \mid \exists v' \in \tilde{y} \,.\, v \text{ is a descendant of } v'\}.$$

We say a leaf node $v \in L$ is *covered* by $\tilde{y}$ if $v \in \text{leaves}(\tilde{y})$. In other words, the size of $\tilde{y}$ is the number of leaf nodes $v$ that are the descendant of some node $v' \in \tilde{y}$. For instance, if $\tilde{y} = \{\text{cat}, \text{dog}\}$, then $\sigma(\tilde{y})$ would be the number of kinds of cats and dogs in the label space. We also assume that the cumulative label probability in (1) has the form

$$\sum_{y \in \mathcal{Y}} g(x, y) \cdot \mathbb{1}(y \in \gamma(\tilde{y})) = \sum_{v \in \text{leaves}(\tilde{y})} p_v.$$

In other words, to obtain the cumulative label probability for $\tilde{y}$, we simply sum the probabilities $p_v$ of the leaf nodes that are covered by $\tilde{y}$. In our image classification example, this value would simply be the sum of all the fine-grained label probabilities that are subsumed by the labels in $\tilde{y}$. With these assumptions, the definition of $h_\tau(x)$ in (1) becomes

$$h_\tau(x) = \arg \min_{\tilde{y} \in \tilde{\mathcal{Y}}} |\text{leaves}(\tilde{y})| \text{ subj. to } \sum_{v \in \text{leaves}(\tilde{y})} p_v \geq \tau.$$

If we additionally unroll the definition of $\tilde{\mathcal{Y}}$, then the objective becomes

$$h_\tau(x) = \arg \min_{\tilde{y} \subseteq V} |\text{leaves}(\tilde{y})| \text{ subj. to } \sum_{v \in \text{leaves}(\tilde{y})} p_v \geq \tau \wedge |\tilde{y}| \leq m, \tag{2}$$

i.e., we need to compute a subset of nodes $\tilde{y}$ of size at most $m$ such that the leaf nodes covered by $\tilde{y}$ have cumulative probability at least $\tau$, while minimizing the number of leaf nodes covered by $\tilde{y}$.

Our framework can also be used to represent qualitatively very different domains from our image classification example. For instance, suppose we want to represent uncertainty in a predicted date; then, we might consider a conjunction of prediction intervals of years—e.g., 1968-1972 or 2010-2012. To do so, each node in $G$ can represent an interval $I = [\ell, u]$, with the children of a node being the intervals $I'$ that are immediately contained in $[\ell, u]$ (i.e., there is no $I''$ in the search space such that $I' \subsetneq I'' \subsetneq I$). Furthermore, we assume that each leaf node of $G$ corresponds to an interval where $\ell = u$, so it represents a single year. Then, $\tilde{y} = \{I_1, ..., I_{m'}\}$ (for some $m' \leq m$) would be a conjunction of intervals, and $\text{leaves}(\tilde{y})$ would be the years contained in at least one of these intervals. We can also handle instances where $G$ varies from one input to another.

We emphasize that our DAG structure is a structure on the space of prediction sets, and can differ from the structure of the label space. In applications where the label space has a tree or DAG structure, we can naturally consider structured prediction sets that conform to this structure, but this is not a requirement. For instance, if the label space has a graph structure, we could still construct prediction sets representing sets of labels, and impose a DAG structure on these prediction sets (e.g., based on set inclusion). Thus, our approach can be flexibly applied to more complex domains.

**Integer programming algorithm.** We describe an integer program to solve (2). For each node $v \in V$, our optimization problem includes two variables $\alpha_v, \beta_v \in \{0, 1\}$. Intuitively, $\alpha_v = 1$ indicates that $v \in \tilde{y}$, and $\beta_v = 1$ indicates that $v$ is covered by $\tilde{y}$ (in general, $v \in V$ is covered by $\tilde{y}$

if there is some $v' \in \tilde{y}$ such that $v$ is a descendant of $v'$). Then, our integer program is

$$\min_{\alpha,\beta} \sum_{v \in L} \beta_v \tag{3}$$

$$\text{subj. to} \sum_{v \in V} \alpha_v \leq m \tag{4}$$

$$\alpha_v \to \beta_v \qquad (\forall v \in V) \tag{5}$$

$$\beta_v \to \beta_{v'} \qquad (\forall (v, v') \in E) \tag{6}$$

$$\beta_{v'} \to \alpha_{v'} \vee \bigvee_{(v,v') \in E} \beta_v \qquad (\forall v' \in V) \tag{7}$$

$$\sum_{v \in L} p_v \cdot \beta_v \geq \tau \tag{8}$$

We have included some Boolean constraints for clarity; in general, a Boolean constraint of the form $\alpha \to \beta$ is equivalent to the linear constraint $\alpha \leq \beta$, and $\alpha \to \beta \vee \beta'$ is equivalent to $\alpha \leq \beta + \beta'$. The objective constraints in our integer program have the following intuition:

- Eq. (3): The objective is to minimize the number of leaf nodes covered by $\tilde{y}$.
- Eq. (4): The prediction set $\tilde{y}$ contains at most $m$ nodes.
- Eq. (5): If $v$ is contained in $\tilde{y}$, then $v$ is covered by $\tilde{y}$.
- Eq. (6): If $v$ is covered by $\tilde{y}$ and $v'$ is a child of $v$, then $v'$ is also covered $\tilde{y}$.
- Eq. (7): If $v'$ is covered by $\tilde{y}$, then either $v'$ is contained in $\tilde{y}$ or at least one parent of $v'$ is covered by $\tilde{y}$.
- Eq. (8): The cumulative probability of leaf nodes covered by $\tilde{y}$ is at least $\tau$.

Given the solution $\alpha^*, \beta^*$ to this integer program, our algorithm returns $\tilde{y} = \{v \in V \mid \alpha_v^* = 1\}$.

## 5 EXPERIMENTS

We empirically validate our approach by demonstrating that it constructs prediction sets that satisfy the desired coverage guarantees while producing reasonably sized prediction sets.[1] We evaluate our approach on five tasks: (i) predicting numbers represented as lists of MNIST digits (LeCun & Cortes, 2010), (ii) ImageNet classification (Deng et al., 2009) with hierarchical label space tasks, (iii) SQuAD question answering (Rajpurkar et al., 2016) where the answer is a year, (iv) Python code generation based on the MBPP dataset (Austin et al., 2021), similar to the task studied in Khakhar et al. (2023), and (v) predicting emotions on the GoEmotions dataset (Demszky et al., 2020).

### 5.1 EXPERIMENTAL SETUP

**MNIST digits.** The goal in this task is to predict a number represented as a list of $k$ MNIST digits (we use $k \in \{2, 3\}$). Each digit is classified using a standard feedforward network $f_\theta$. We consider prediction sets inspired by significant figures for representing measurements—namely, a prediction set is represented by a confident prediction in the first $h \leq k$ digits, and uncertain in the remaining ones. In more detail, a prediction set has the form $\tilde{y} = [d_1, ..., d_k]$, where $d_i \in \{0, 1, ..., 9, \varnothing\}$ for each $i \in [k]$, and where $d_i = \varnothing$ implies that $d_{i'} = \varnothing$ for all $i' \geq i$. Then, we have

$$\gamma(\tilde{y}) = \left\{ \sum_{i=1}^{k} d_i' \cdot 10^{k-i} \;\middle|\; d_i' \in \gamma(d_i) \right\} \qquad \text{where} \qquad \gamma(d) = \begin{cases} \{d\} & \text{if } d \neq \varnothing \\ \{0, 1, ..., 9\} & \text{otherwise.} \end{cases}$$

For example, $[1, 2, \varnothing]$ represents the prediction set $\{120, 121, ..., 129\}$. According to our optimization problem setup, the prediction set consists of at most $m$ intervals, which together form a non-continuous digit interval as the final prediction. Finally, the DAG is constructed by having

---

[1]The implementation is available at
https://github.com/botong516/Conformal-Structured-Prediction.

$[d_1, ..., d_i, \varnothing, \varnothing, ..., \varnothing] \rightarrow [d_1, ..., d_i, d_{i+1}, \varnothing, ..., \varnothing]$ for all $d_1, ..., d_i, d_{i+1} \neq \varnothing$. In other words, one node is the child of another if it predicts the same prefix and exactly one additional digit. A leaf node of this DAG is a sequence of $k$ digits $[d_1, ..., d_k]$; we associate with this leaf node with the probability $\prod_{i=1}^{k} f_\theta(d_i \mid x_i)$, where $[x_1, ..., x_k]$ is the sequence of input images and $f_\theta(d_i \mid x_i)$ is the predicted probability that $x_i$ is an image of digit $d_i$ according to $f_\theta$.

**Image classification with hierarchical labels.** We consider image classification using ResNet-50 (He et al., 2015) on ImageNet (Deng et al., 2009). In this domain, the DAG is a tree; we take the original 1000 ImageNet labels to be leaves of the tree, and a standard set of coarse-grained labels (e.g., animal, living thing, etc.) as the internal nodes. We include an edge $(v, v') \in E$ if $v$ is an immediate hypernym of $v'$ in the WordNet lexical database (Miller, 1994). A prediction set $\tilde{y} = \{\ell_1, ..., \ell_{m'}\}$ is a set of $m' \leq m$ internal nodes, representing the set of fine-grained labels corresponding to leaf nodes that are descendants of some $\ell \in \tilde{y}$ (e.g., the prediction set might be "dog or cat"). We associate with each leaf node the probability $f_\theta(\ell \mid x)$, where $\ell$ is the fine-grained label associated with the leaf node, $x$ is the input image, and $f_\theta$ is the pretrained ResNet-50 model.

**Question answering about dates.** We consider a question answering task based on the Stanford Question Answering Dataset (SQuAD) (Rajpurkar et al., 2016), but focusing on questions where the answer is a year in the range from 1970 to 2020. The DAG is constructed as the set of intervals $[\ell, u]$, where $\ell, u \in \{1970, 1971, \ldots, 2020\}$. We have edge $[\ell, u] \rightarrow [\ell', u']$ if $[\ell', u'] \subsetneq [\ell, u]$, and there is no interval $[\ell'', u'']$ such that $[\ell', u'] \subsetneq [\ell'', u''] \subsetneq [\ell, u]$. Each leaf node is a year $[\ell, u]$ where $\ell = u$; we associate it with the probability of the answer being $\ell$ as predicted by the Llama-3.1-70B-Instruct model (Dubey et al., 2024)—i.e., $f_\theta(\ell \mid x)$, where $x$ is the question, $f_\theta$ is the Llama model, and $f_\theta(\ell \mid x)$ is the probability of the sequence of tokens representing the year $\ell$ with a standard question answering prompt asking the model for the answer to question $x$. We provide additional details on the dataset, prompt, and the DAG structure in Appendix A.1.

**Python code generation.** We investigate the problem of code generation in Python using the MBPP dataset (Austin et al., 2021). In our experiments, we provided the gpt-4o-mini model with a natural language prompt along with $k$ lines of code from the original ground truth program in the dataset, instructing the model to complete the program to solve the prompt. In this context, the DAG structure is defined by the AST parsed from the generated program. We assign each node a probability equal to the sum of the probabilities of the tokens contained with that node.

**Emotion label prediction for text.** Finally, we use the GoEmotions dataset (Demszky et al., 2020), which consists of 27 emotion categories annotated on 58,000 English Reddit comments, to demonstrate the application of our framework to predict the emotion labels in a given piece of text. Similar to the ImageNet task, the DAG in this domain is also a tree, following the structure proposed in the GoEmotions dataset (Demszky et al., 2020). We use the provided set of concrete emotion labels (e.g., amusement, fear, grief, etc.) as the leaves of the tree. The definitions of prediction sets and leaf node probabilities are consistent with those from the ImageNet task, with $f_\theta$ being a pretrained RoBERTa base model (Sam Lowe, 2024).

**Hyperparameters.** We use $m \in \{1, 2, 4, 8\}$ (default of $m = 4$), $\epsilon \in \{0.05, 0.1, 0.15, 0.2\}$ (default of $\epsilon = 0.1$), and $\delta \in \{0.1, 0.01, 0.001\}$ (default of $\delta = 0.01$).

**Baseline.** We compare our approach with a baseline strategy adapted from Khakhar et al. (2023). While the strategy proposed is specialized to the code domain, we generalize it to apply to arbitrary DAG structures. Unlike our approach, this baseline leverages existing PAC prediction set algorithms, which require that the monotonicity assumption holds. Thus, their algorithm restricts the structure of the prediction sets across different values of $\tau$ to enforce monotonicity. In contrast, our approach proves a novel conformal prediction bound (Theorem 3.2) to avoid the need for monotonicity.

**Metrics.** We report empirical coverage rate and average prediction set size on a held-out test set:

$$\text{Coverage Rate} = \frac{1}{|Z_{\text{test}}|} \sum_{(x, y^*) \in Z_{\text{test}}} \mathbb{1}(y^* \in \gamma(\tilde{y}))$$

$$\text{Average Prediction Set Size} = \frac{1}{|Z_{\text{test}}|} \sum_{(x, y^*) \in Z_{\text{test}}} \sigma(\tilde{y}).$$

Our goal is for the coverage rate to exceed $\epsilon$ while minimizing average prediction set size. We show averages and standard deviations over 5 runs.

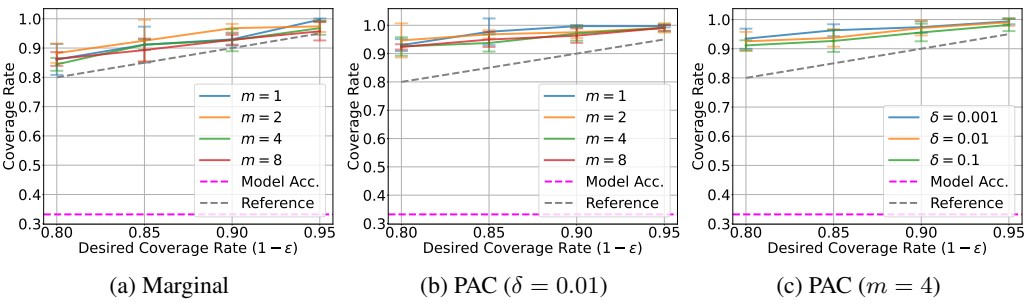

(a) Marginal          (b) PAC ($\delta = 0.01$)          (c) PAC ($m = 4$)

Figure 2: Prediction set coverage rates for the question answering task, for (a) marginal guarantee, (b) PAC guarantee with fixed $\delta$ and varying $m$, and (c) PAC guarantee with fixed $m$ and varying $\delta$.

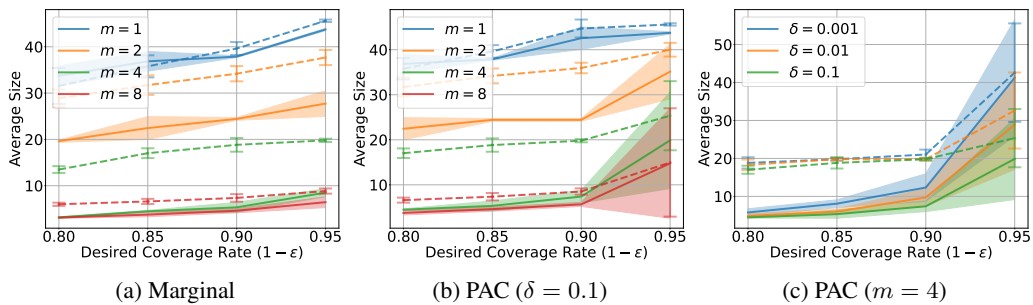

(a) Marginal          (b) PAC ($\delta = 0.1$)          (c) PAC ($m = 4$)

Figure 3: Prediction set sizes for the question answering task, with the baseline represented by dashed lines, for (a) marginal guarantee, (b) PAC guarantee with fixed $\delta$ and varying $m$, and (c) PAC guarantee with fixed $m$ and varying $\delta$.

## 5.2 RESULTS

Here, we show results only for the question answering task; results for the MNIST, ImageNet, code generation, and GoEmotions tasks can be found in Appendix A.4, and exhibit similar trends.

**Coverage guarantees.** First, we study the coverage rates achieved by our approach. Results are shown in Figure 2; Figure 2a shows results with the marginal guarantee for different $m$ values across the given error levels $\epsilon$, Figure 2b shows results with the PAC guarantee for different $m$ and fixed $\delta$, and Figure 2c shows results with the PAC guarantee for different $\delta$ and fixed $m$; coverage rates for the baseline are shown in Figure 11 in Appendix A.4. In general, for both our algorithm and for the baseline, the empirical coverage rates are above the desired coverage level (i.e., the "Reference" line). As expected, as $\epsilon$ increases, the coverage rate tends to decrease (but remains above the desired coverage rate); $m$ and $\delta$ do not significantly affect coverage.

For the marginal guarantee, mean coverage rates remain close to the desired rate, while for the PAC guarantee, coverage for all values holds within one standard deviation. Note that marginal guarantees are on average over both the training set and the new examples; thus, the average coverage across different random seeds is above the desired coverage level, but any individual random seed may fall above or below this level. In contrast, PAC prediction sets hold with high probability over the training set; thus, the coverage is above the desired level for almost all random seeds. Intuitively, PAC guarantees are more conservative than marginal guarantees but provide greater reliability. In domains where it is critical for the deployed model to satisfy the coverage guarantee, PAC prediction sets should be used; otherwise, marginal guarantees may suffice.

**Prediction set size.** Next, we study the average size of the prediction sets constructed using our approach in terms of number of concrete labels and compare it to the prediction sets obtained using the baseline. Results are shown in Figure 3; Figure 3a shows results with the marginal guarantee for different $m$ values across the given error levels $\epsilon$, Figure 3b shows results with the PAC guarantee for different $m$ and fixed $\delta$, and Figure 3c shows results with the PAC guarantee for different $\delta$ and fixed $m$. Results for the baseline are represented by dashed lines. As expected, prediction set size

|          | $\epsilon = 0.05$ | $\epsilon = 0.1$ | $\epsilon = 0.2$ |
|----------|-------------------|------------------|------------------|
| $m = 1$  | $\{[1979, 2019]\}$ | $\{[1979, 2019]\}$ | $\{[1987, 2020]\}$ |
| $m = 2$  | $\left\{\begin{array}{l}[1979, 1980], \\ [1997, 2019]\end{array}\right\}$ | $\left\{\begin{array}{l}[1979], \\ [1997, 2019]\end{array}\right\}$ | $\left\{\begin{array}{l}[1996, 2001], \\ [2007, 2020]\end{array}\right\}$ |
| $m = 4$  | $\left\{\begin{array}{l}[1979, 1980], \\ [1997], \\ [2007], \\ [2015, 2019]\end{array}\right\}$ | $\left\{\begin{array}{l}[1979], \\ [1997], \\ [2007], \\ [2015, 2019]\end{array}\right\}$ | $\left\{\begin{array}{l}[1997], \\ [2015], \\ [2016, 2017], \\ [2019]\end{array}\right\}$ |

Table 1: Structured prediction sets for the question, *"When was 7 Lincoln Square completed?"* with the ground truth answer "1979". Intervals in red contain the ground truth year.

decreases as $\epsilon$ and $\delta$ increase and as $m$ increases; its dependence on $\delta$ is much less pronounced than its dependence on $\epsilon$ and $m$. Our approach outperforms the baseline in terms of prediction set size for almost every parameter setting, and significantly so for $m = 2$ and $m = 4$.

In general, the sensitivity of the hyperparameter $m$ depends on the DAG structure and the performance of the underlying model. For the question answering task, the decrease in size with $m$ is especially significant, likely due to the low accuracy of the underlying model ($\approx 33.2\%$). Since all nodes tend to have similar probability masses, usually without any dominant ones, changes in $m$ can significantly affect the nodes selected when $m$ is small (e.g., 1 or 2). As $m$ becomes larger (e.g., 4 or 8), the algorithm gains greater flexibility in selecting which nodes to include in the prediction set; thus, the prediction set size typically becomes less sensitive to $m$ as it becomes larger.

**Qualitative examples.** We provide a qualitative example from the question answering task to show: (i) how structured sets differ from standard conformal prediction sets, and (ii) how hyperparameters influence the sets. In the example presented in Table 1, standard conformal prediction yields a prediction set containing six years—$\{1979, 1997, 2007, 2016, 2017, 2019\}$—spanning a broad range of possibilities. We then construct structured prediction sets with $m \in \{1, 2, 4\}$ and $\epsilon \in \{0.05, 0.1, 0.2\}$. Note that increasing $m$ increases the number of intervals but reduces the number of labels in the concrete set. Also, for a fixed $m$, increasing $\epsilon$ generally produces fewer and narrower intervals, albeit at the expense of higher miscoverage rates. In this example, $m = 4$ provides a good tradeoff, summarizing $\{2016, 2017, 2019\}$ into the interval $[2015, 2019]$ while preserving the singleton years 1979, 1997, and 2007. See Appendix A.3 for additional examples.

In general, the choice of $m$ depends on the needs of the given application domain. It governs the trade-off between the interpretability and granularity of the resulting prediction sets. In particular, larger values of $m$ allow more labels to be included in the set, often capturing finer-grained categories such as "attire" or "Blenheim spaniel". Conversely, smaller values of $m$ result in sets with fewer or coarse-grained labels such as "artifact" or "dog", which can be more interpretable for users. We suggest that practitioners try different values of $m$, and manually examine the resulting prediction sets to determine which choices offer the best tradeoff between interpretability and coverage.

**Computational cost.** We also evaluate the running time of our approach compared to our baseline. We find that (i) our approach outperforms the baseline in terms of running time for most domains, (ii) increasing the DAG size does not necessarily lead to higher computation time, (iii) running time usually remains consistent across hyperparameters, (iv) the overall running time is smaller for fixed DAGs where we can leverage warm-starting; see Appendix A.2 for details.

## 6 CONCLUSION

We have proposed a novel algorithm for conformal structured prediction that constructs structured prediction sets that satisfy either a marginal or PAC coverage guarantee. Our algorithm enables uncertainty quantification in settings where the label space cannot be represented simply by a collection of regression and/or classification outputs. Furthermore, we have demonstrated how our approach can be applied to domains where the structured prediction sets are defined by a DAG, which includes settings such as hierarchical labels and text generation. Finally, in our empirical evaluation, we have demonstrated that our algorithm can construct reasonable prediction sets that satisfy the desired coverage guarantee across several application domains.

ACKNOWLEDGEMENTS

This work was supported by NSF Award CCF-1917852 and ARO Award W911NF20-1-0080.

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

## A    APPENDIX

### A.1    EXPERIMENTAL DETAILS ON QUESTION ANSWERING TASK

The SQuAD dataset contains 691 questions where the set of possible answers include at least one that is a four-digit number. From this subset, we conducted experiments on questions with answers between 1970 and 2020, resulting in a total of 262 examples. Each example is a *(question, context, answer)* triplet, where the context provides background information to answer the question. We used a two-shot prompting technique to obtain the log probability from the Llama-3.1-70B-Instruct model (Dubey et al., 2024). Figure 4 shows the prompt template used, where **{question}** and **{context}** are replaced based on the current example. Then, for each year in [1970, 2020], we compute the model's probability of generating **{year}** as a response to the **{question}** and **{context}**. Finally, Figure 5 illustrates the DAG used in this task.

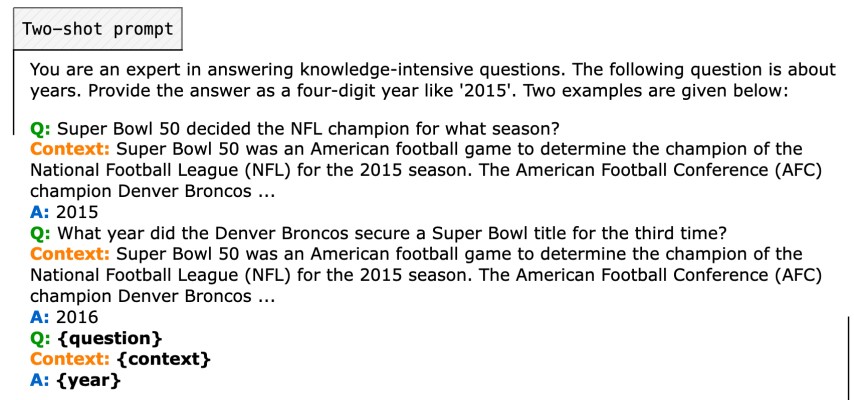

Figure 4:  The two-shot prompt used for our question answering task; here, **{question}** and **{context}** are replaced with the corresponding data from each example in our dataset, and **{year}** is replaced with each year between 1970 and 2020.

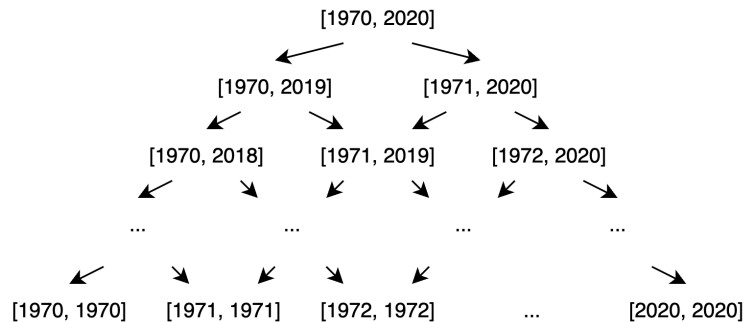

Figure 5: Illustration of the DAG structure for our question answering task.

### A.2    COMPUTATIONAL COST AND SCALABILITY

Our method leverages Integer Programming (IP) to select nodes as prediction sets on DAGs, which can sometimes be computationally intensive, particularly as the size of the DAG increases. We evaluate the running time and scalability of this approach, including a comparison to our baseline adapted from Khakhar et al. (2023). Figure 6 shows the average time required to solve each IP problem as a function of $m$, for each of our application domains. As can be seen, both the code generation and GoEmotions tasks have very fast solve times due to their small tree sizes. As shown in the top right subplot of each plot, MBPP is still significantly faster than GoEmotions. The MBPP dataset consists of entry-level Python programs; therefore, the resulting ASTs are usually very small. The GoEmotions tree structure is also relatively simple, consisting of 8 layers, 27 leaves, and 52

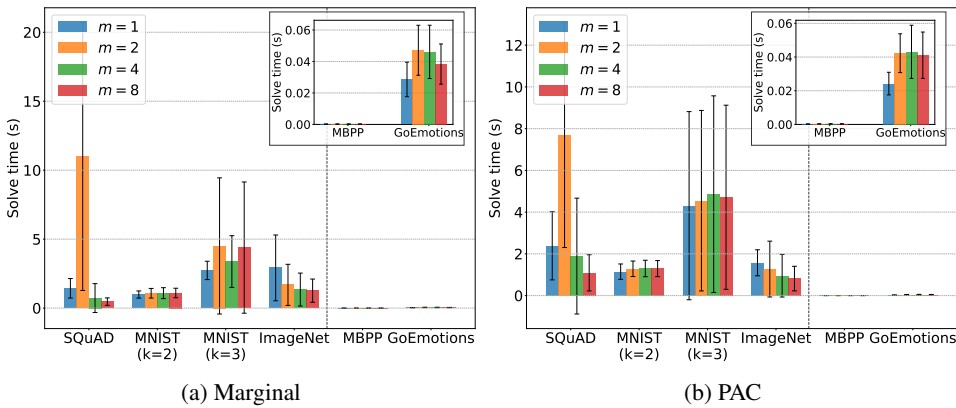

Figure 6: Average time (second) required to solve each Integer Programming (IP) problem in the test set for each of the five domains in our experiments, as $m$ varies, for (a) marginal guarantee and (b) PAC guarantee.

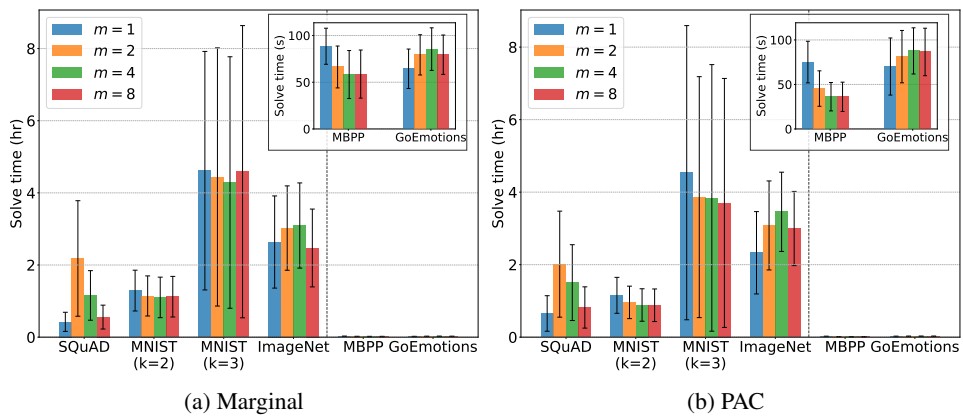

Figure 7: Average time (hour) required to estimate $\tau$ for each of the five domains in our experiments, as $m$ varies, for (a) marginal guarantee and (b) PAC guarantee.

nodes. The 2-digit MNIST and ImageNet problems also have relatively fast solve times. The 2-digit MNIST tree has 3 layers, 100 leaves, and 111 nodes, whereas the ImageNet tree is a lot larger, with 18 layers, 1000 leaves, and 1816 nodes. Thus, increasing the DAG size may not increase computation time.

When extending to the 3-digit MNIST problem, the computation time for the IP becomes slower, especially for the PAC guarantee. The 3-digit MNIST tree has 4 layers, 1000 leaves, and 1111 nodes, which significantly increases the number of nodes compared to the 2-digit MNIST tree and notably increases the tree density compared to the ImageNet tree. Thus, the computation for the IP may become intensive as the density of the DAG scales up. A practical strategy to alleviate this computational burden is to simplify the DAG by removing some internal nodes while preserving the overall hierarchy. In particular, if a node $v$ is removed, its parent nodes become the parents of each of $v$'s children. This approach allows us to maintain the structured prediction set property while improving computational efficiency, though it may affect interpretability of the resulting structured prediction sets.

The structure for the question answering task is not a tree but a DAG, where a single node can have multiple parents. It has 51 layers, 51 leaves, and 650 nodes, making it relatively sparse compared to structures for the other four domains. In this case, there is a drastic increase in running time when $m = 2$. As we leverage an off-the-shelf optimizer, the process of node selection largely occurs under the hood. Consequently, there may be cases where the runtime is unusually high on certain

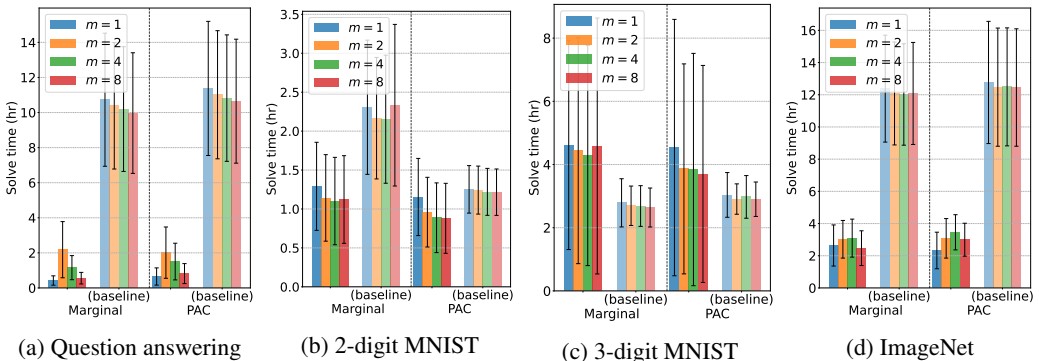

| (a) Question answering | (b) 2-digit MNIST | (c) 3-digit MNIST | (d) ImageNet |

Figure 8: Average time (hour) required to complete each of the four tasks—(a) question answering, (b) 2-digit MNIST, (c) 3-digit MNIST, and (d) ImageNet—with nontrivial solve times (as shown in Figure 6 and Figure 7), along with their respective baselines, as $m$ varies, for both marginal and PAC guarantees.

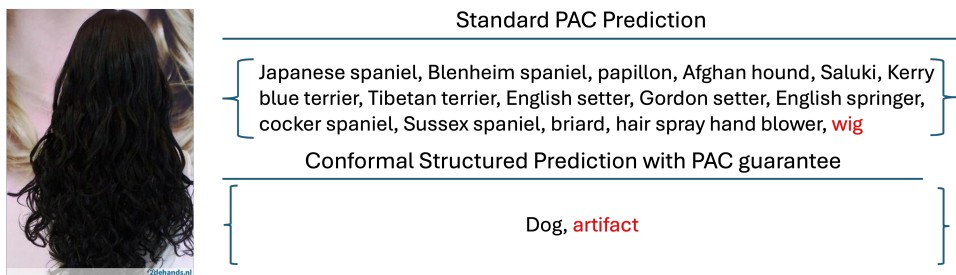

Figure 9: Comparison of prediction sets generated by conformal structured prediction under PAC guarantee and the standard PAC prediction algorithm (Vovk, 2012; Park et al., 2020). For both methods, the error level is $0.05$ and the confidence level is $0.99$ ($\delta = 0.01$).

DAGs. Selecting a different $m$ can help mitigate this issue. As shown in the plot, the running time for a given type of problem remains consistent overall across different hyperparameters.

It is important to note that the runtime for solving each individual IP problem does not always reflect the overall runtime of the entire framework. When it comes to solving sequences of IP problems (such as during $\tau$ estimation), certain problem setups can largely benefit from reduced runtime through the chosen optimizer's built-in techniques. For instance, as shown in Figure 7, the Python code generation problem loses its advantage of extremely fast IP solve times during the $\tau$ estimation phase, likely because the DAG structure (i.e. AST tree) varies across samples. As a result, additional computation time is required to build each DAG, and furthermore we cannot use warm-starting.

Finally, our approach significantly outperforms the baseline in most domains. Figure 8 shows the average time required to complete the four tasks with nontrivial solve times (i.e., question answering, 2-digit MNIST, 3-digit MNIST, and ImageNet). For the question answering and ImageNet task, our approach is significantly faster than the baseline. In the 2-digit MNIST problem, we outperform the baseline with stronger improvements under the marginal guarantee. However, for the 3-digit MNIST problem, our approach is slightly slower than the baseline. The faster running time of our approach is likely due to having fewer restrictions on the structure of the prediction sets across different $\tau$.

### A.3 ADDITIONAL QUALITATIVE EXAMPLES

**Comparison with standard PAC prediction sets.** Figure 9 compares our approach to the standard PAC prediction set algorithm (Vovk, 2012; Park et al., 2020), with $\epsilon = 0.05$ and $\delta = 0.01$. As can be seen, the standard prediction set is much larger than the one constructed by our algorithm. The large prediction set includes labels from very distant categories, making it harder to interpret compared to having just two coarse-grained labels that summarize the labels. In our structured prediction set, the

|  | $\epsilon = 0.05$ | $\epsilon = 0.1$ | $\epsilon = 0.2$ |
|---|---|---|---|
| $m = 1$ | {whole} | {artifact} | {hairpiece} |
| $m = 2$ | {dog, artifact} | {toy spaniel, attire} | {wig} |
| $m = 4$ | {toy dog, clothing, hunting dog} | {Blenheim spaniel, attire, English setter} | {wig} |

Table 2: Structured prediction sets for the image in Figure 9. Labels in red contain the ground class.

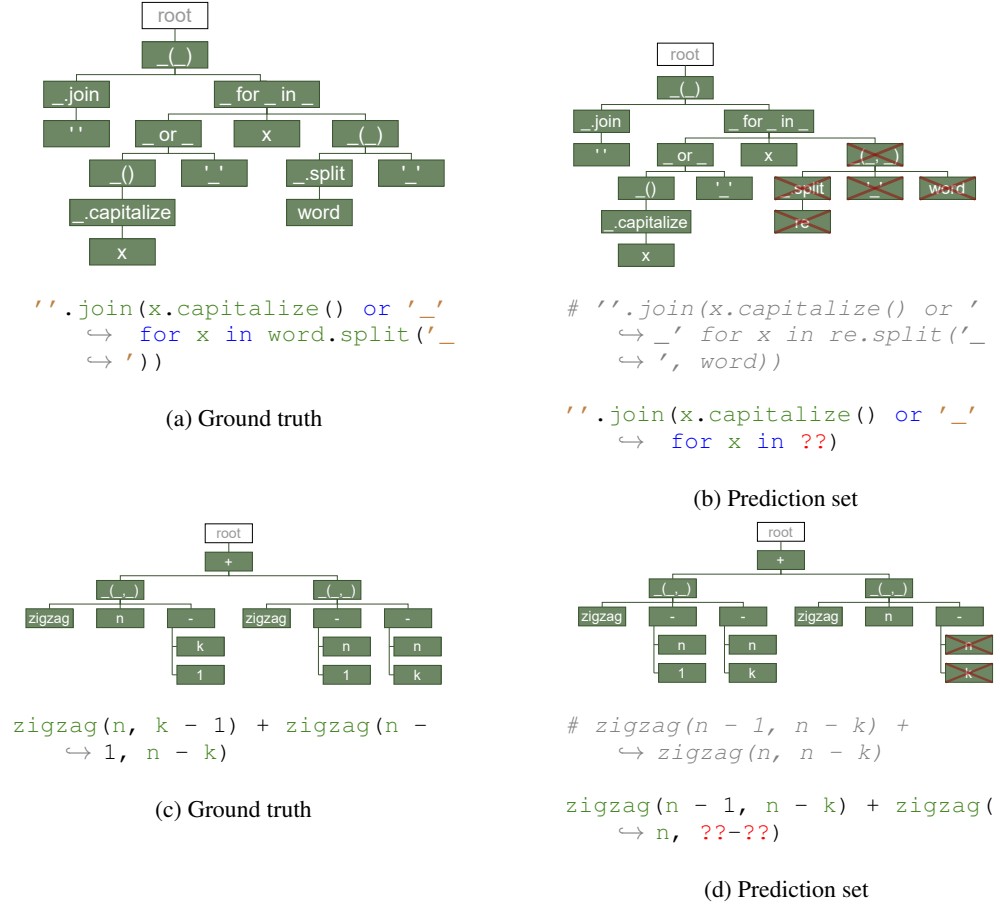

Figure 10: Structured prediction sets for the code generation domain. We show (a,c) the ground truth AST, and (b,d) the predicted AST (nodes removed by our algorithm to obtain the prediction set are crossed out in red). The corresponding code is shown below (for (b,d), commented code is the original prediction). The hyperparameters used are $\epsilon = 0.2, m = 1$ for (b) and $\epsilon = 0.15, m = 2$ for (d).

different dog breeds have been summarized as simply "dog", and the different man-made artifacts have been summarized as "artifact". The fact that the labels are quite coarse reflects the inherent uncertainty in the prediction for this image.

**Impact of hyperparameters.** In Table 2, we show additional prediction sets for the same image as in Figure 9 to show how hyperparameters influence the prediction sets. Similar to our findings in Table 1, for smaller $m$, our prediction sets contain fewer labels, making them more interpretable, while prediction sets with larger $m$ contain more fine-grained labels. Also, with higher $\epsilon$, our algorithm is allowed to make more errors, resulting in much smaller prediction sets.

**Example structured prediction sets in code domain.** We show two examples of structured prediction sets in the code generation domain in Figure 10. In each example, the AST on the left represents the ground truth program, with its corresponding code generation shown directly below. The AST on the right, with certain nodes removed (crossed in red), represents a structured prediction set for it, where the partial code is shown under the figure with the complete code prediction provided in the comment. The removed portions are denoted as ?? in the generated code to indicate uncertainty. Compared to traditional prediction sets consisting of multiple complete programs, the structured prediction sets improve interpretability by explicitly preserving the confident portions of the code while leaving the uncertain parts unspecified. Note that after removing the uncertain components, the remaining partial programs each contain the respective ground truth program.

## A.4 ADDITIONAL QUANTITATIVE RESULTS

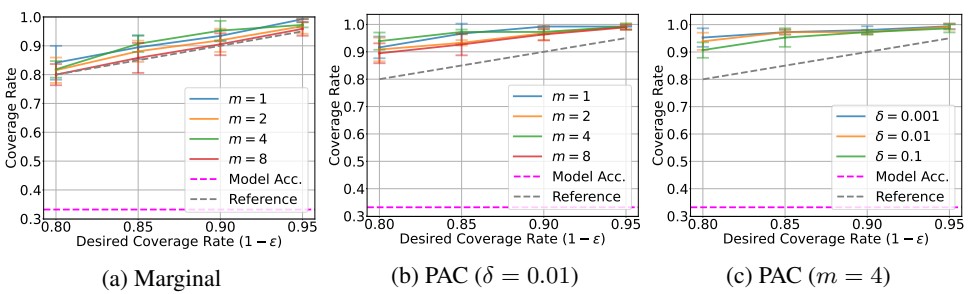

Figure 11: Coverage rates for the **question answering task using the baseline**, for (a) marginal guarantee, (b) PAC guarantee with fixed $\delta$ and varying $m$, and (c) PAC guarantee with fixed $m$ and varying $\delta$.

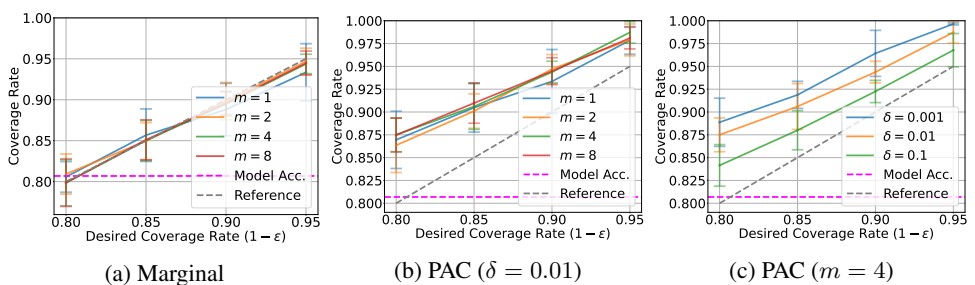

Figure 12: Coverage rates for the **2-digit MNIST** task with $n = 200$, for (a) marginal guarantee, (b) PAC guarantee with fixed $\delta$ and varying $m$, and (c) PAC guarantee with fixed $m$ and varying $\delta$.

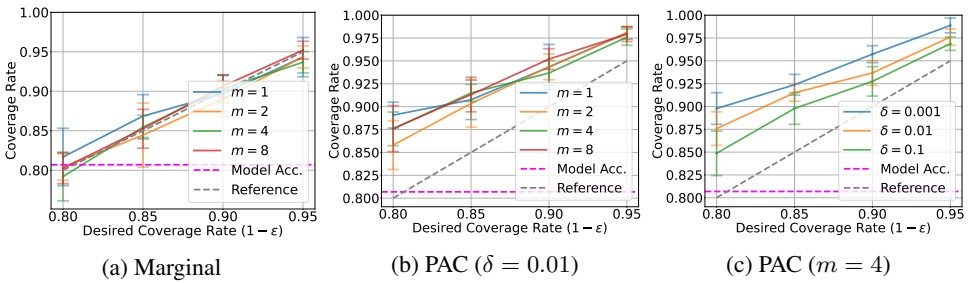

Figure 13: Coverage rates for the **2-digit MNIST task using the baseline** with $n = 200$, for (a) marginal guarantee, (b) PAC guarantee with fixed $\delta$ and varying $m$, and (c) PAC guarantee with fixed $m$ and varying $\delta$.

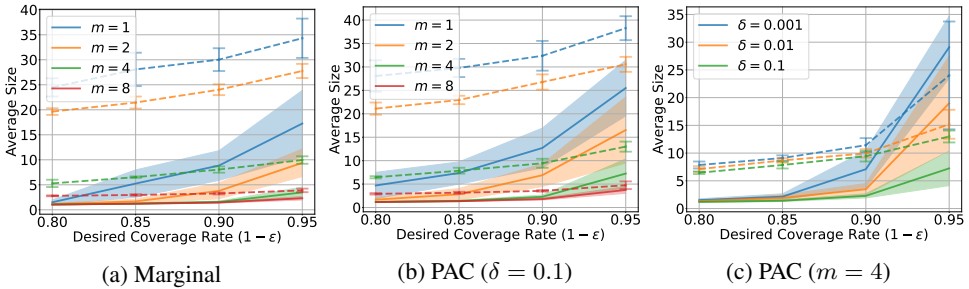

Figure 14: Prediction set sizes for the **2-digit MNIST** task ($n = 200$), with the baseline represented by dashed lines, for (a) marginal guarantee, (b) PAC guarantee with fixed $\delta$ and varying $m$, and (c) PAC guarantee with fixed $m$ and varying $\delta$.

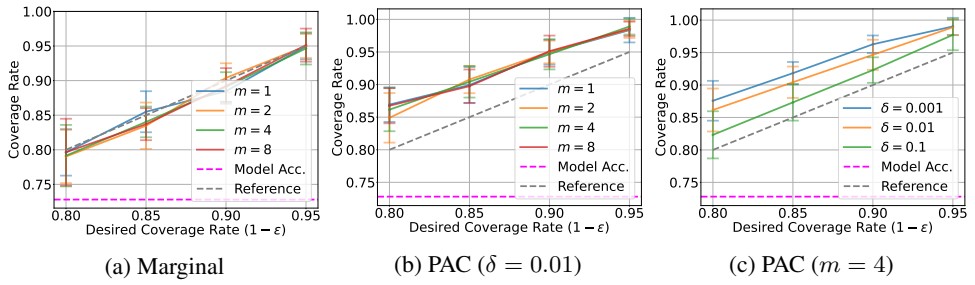

Figure 15: Coverage rates for the **3-digit MNIST** task with $n = 200$, for (a) marginal guarantee, (b) PAC guarantee with fixed $\delta$ and varying $m$, and (c) PAC guarantee with fixed $m$ and varying $\delta$.

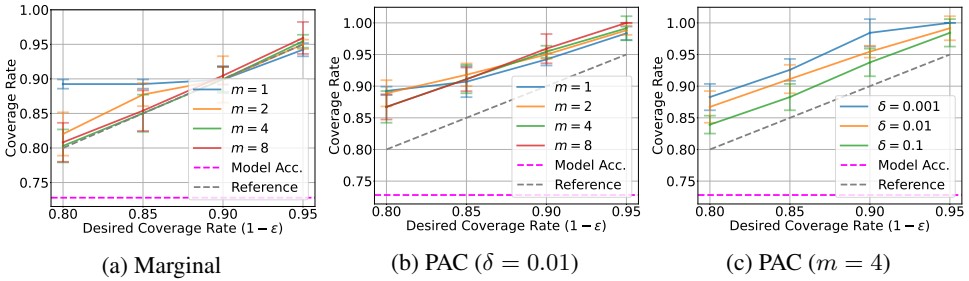

Figure 16: Coverage rates for the **3-digit MNIST task using the baseline** with $n = 200$, for (a) marginal guarantee, (b) PAC guarantee with fixed $\delta$ and varying $m$, and (c) PAC guarantee with fixed $m$ and varying $\delta$.

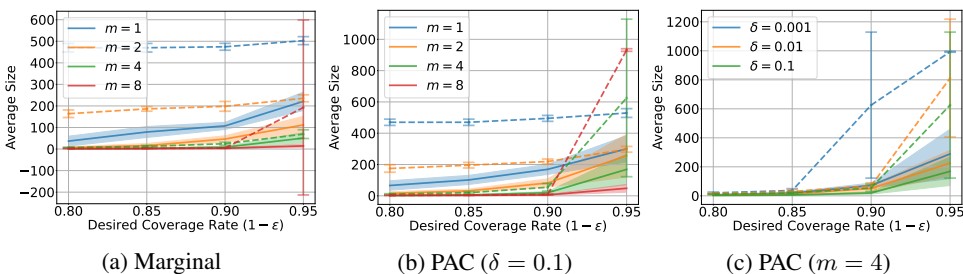

Figure 17: Prediction set sizes for the **3-digit MNIST** task ($n = 200$), with the baseline represented by dashed lines, for (a) marginal guarantee, (b) PAC guarantee with fixed $\delta$ and varying $m$, and (c) PAC guarantee with fixed $m$ and varying $\delta$.

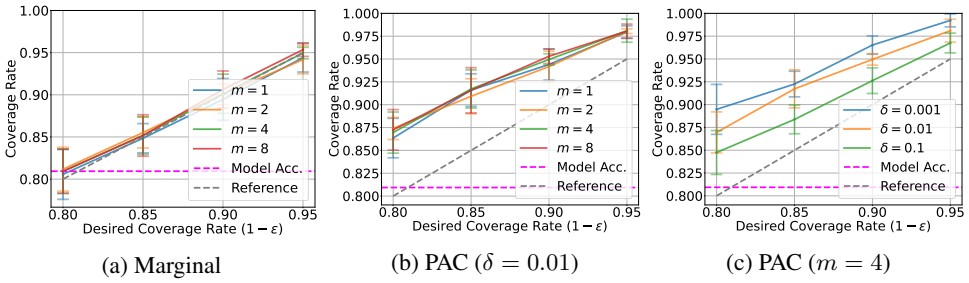

Figure 18: Coverage rates for the **ImageNet** task with $n = 200$, for (a) marginal guarantee, (b) PAC guarantee with fixed $\delta$ and varying $m$, and (c) PAC guarantee with fixed $m$ and varying $\delta$.

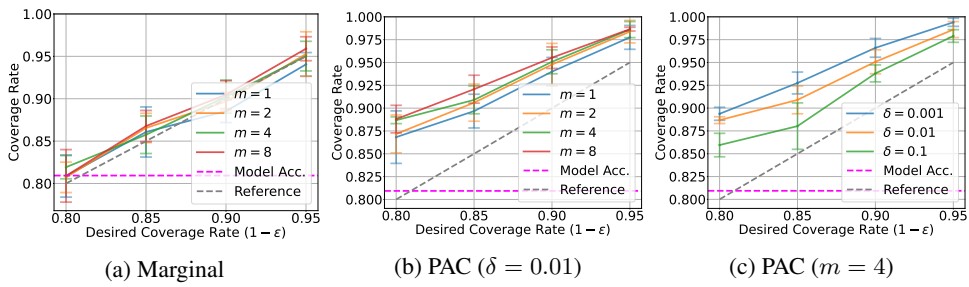

Figure 19: Coverage rates for the **ImageNet task using the baseline** with $n = 200$, for (a) marginal guarantee, (b) PAC guarantee with fixed $\delta$ and varying $m$, and (c) PAC guarantee with fixed $m$ and varying $\delta$.

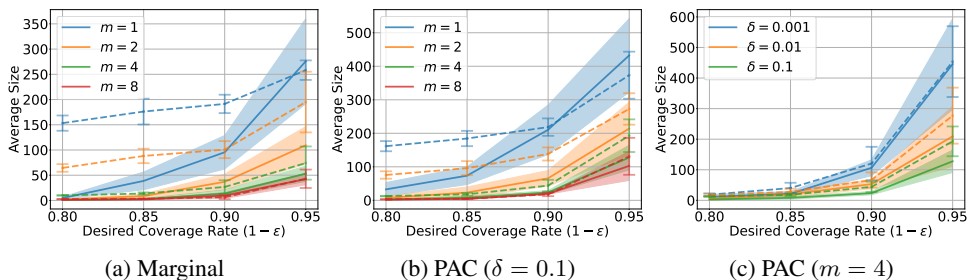

Figure 20: Prediction set sizes for the **ImageNet** task ($n = 200$), with the baseline represented by dashed lines, for (a) marginal guarantee, (b) PAC guarantee with fixed $\delta$ and varying $m$, and (c) PAC guarantee with fixed $m$ and varying $\delta$.

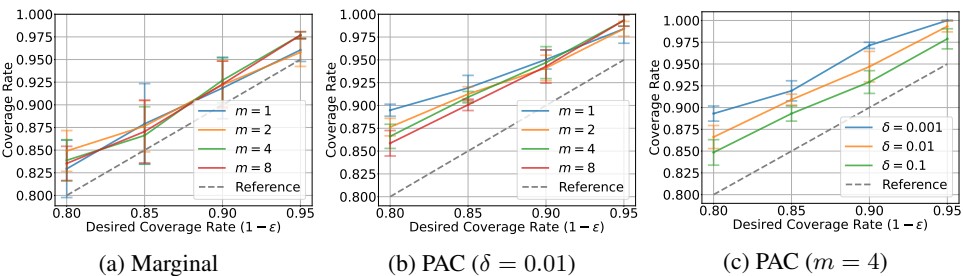

Figure 21: Coverage rates for the **Python code generation** task with $n = 233$, for (a) marginal guarantee, (b) PAC guarantee with fixed $\delta$ and varying $m$, and (c) PAC guarantee with fixed $m$ and varying $\delta$.

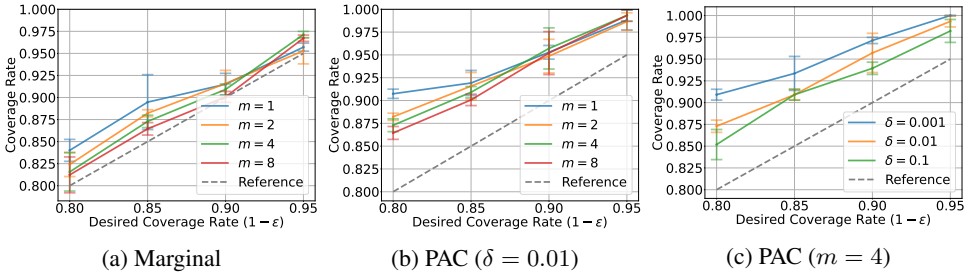

(a) Marginal  (b) PAC ($\delta = 0.01$)  (c) PAC ($m = 4$)

Figure 22: Coverage rates for the **Python code generation task using the baseline** with $n = 233$, for (a) marginal guarantee, (b) PAC guarantee with fixed $\delta$ and varying $m$, and (c) PAC guarantee with fixed $m$ and varying $\delta$.

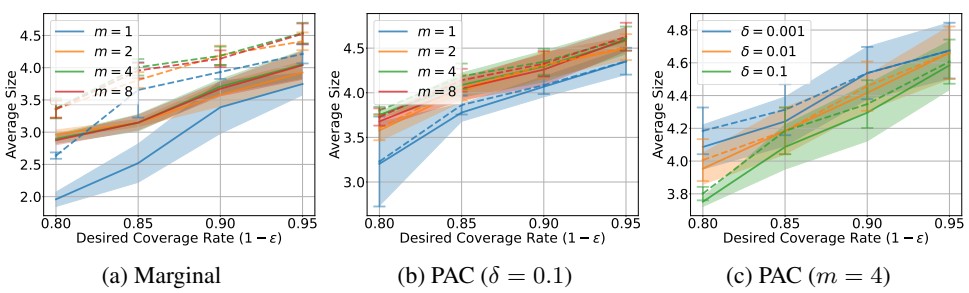

(a) Marginal  (b) PAC ($\delta = 0.1$)  (c) PAC ($m = 4$)

Figure 23: Prediction set sizes for the **Python code generation** task ($n = 233$), with the baseline represented by dashed lines, for (a) marginal guarantee, (b) PAC guarantee with fixed $\delta$ and varying $m$, and (c) PAC guarantee with fixed $m$ and varying $\delta$.

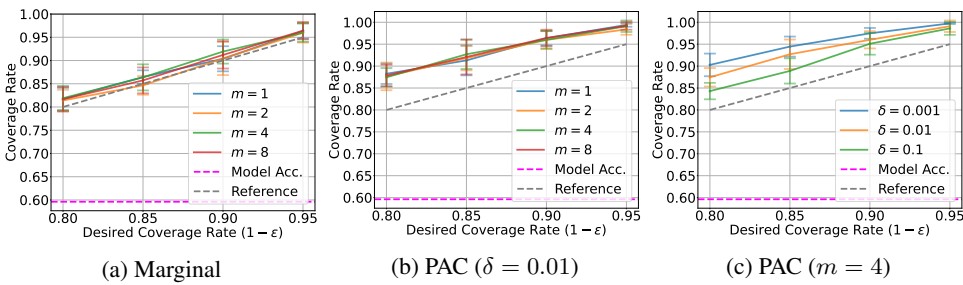

(a) Marginal  (b) PAC ($\delta = 0.01$)  (c) PAC ($m = 4$)

Figure 24: Coverage rates for the **GoEmotions** task with $n = 200$, for (a) marginal guarantee, (b) PAC guarantee with fixed $\delta$ and varying $m$, and (c) PAC guarantee with fixed $m$ and varying $\delta$.

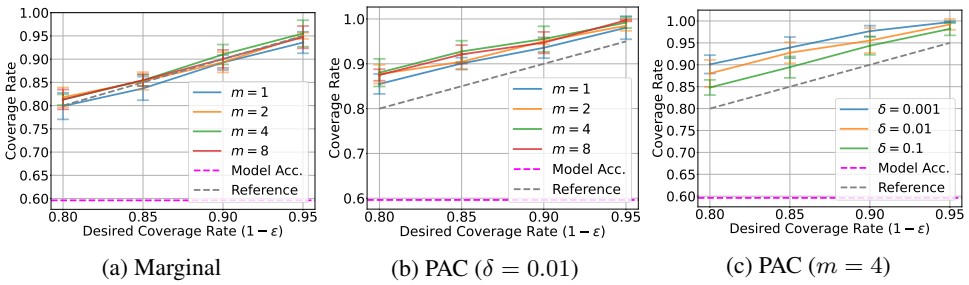

(a) Marginal  (b) PAC ($\delta = 0.01$)  (c) PAC ($m = 4$)

Figure 25: Coverage rates for the **GoEmotions task using the baseline** with $n = 200$, for (a) marginal guarantee, (b) PAC guarantee with fixed $\delta$ and varying $m$, and (c) PAC guarantee with fixed $m$ and varying $\delta$.

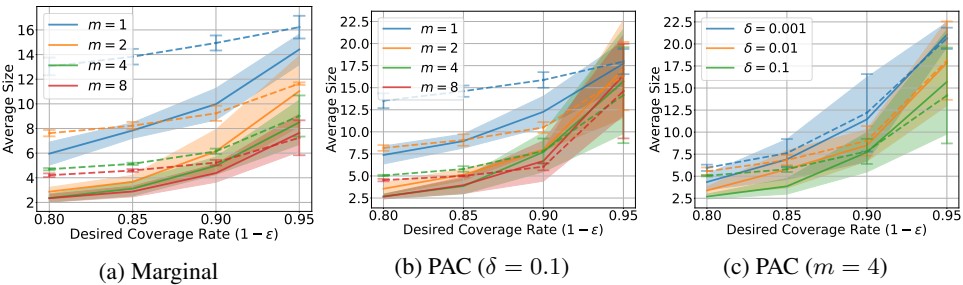

Figure 26: Prediction set sizes for the **GoEmotions** task ($n = 200$), with the baseline represented by dashed lines, for (a) marginal guarantee, (b) PAC guarantee with fixed $\delta$ and varying $m$, and (c) PAC guarantee with fixed $m$ and varying $\delta$.

