# OpenReview forum: "Conformal Structured Prediction"
_ICLR.cc/2025/Workshop/BuildingTrust — BuildingTrust_

### Official Review · Reviewer_cmaJ · 2025-02-23

**Rating:** 6
**Confidence:** 3

**Review:**

conformal prediction methods to more complex structured prediction tasks, such as code generation and question answering. The framework generates structured prediction sets, offering a compact and interpretable representation of uncertainty. Specifically, it models the prediction sets as subgraphs of a directed acyclic graph (DAG), which is particularly useful in tasks like code generation, where the prediction set can be represented as a partially completed program. The approach ensures reliable coverage guarantees, either marginal or PAC (probably approximately correct), and applies it to domains such as the SQuAD question answering dataset and Python code generation tasks. Experimental results demonstrate that the framework efficiently constructs small, interpretable prediction sets while maintaining high coverage rates and outperforming existing methods in terms of prediction set size.

Strengths
- The framework provides robust coverage guarantees, ensuring high reliability in structured prediction tasks with interpretable uncertainty.
- By representing predictions as subgraphs of a directed acyclic graph (DAG), the method delivers smaller and more interpretable prediction sets, improving clarity.
- The approach outperforms existing methods in terms of prediction set size while maintaining high coverage rates, demonstrating its effectiveness in tasks like code generation and question answering.

Weaknesses
- The method's use of directed acyclic graphs (DAGs) for prediction sets introduces additional complexity in both model training and implementation.
- While effective for structured prediction tasks, the approach may face challenges when applied to other types of prediction tasks outside the structured domain.
- Generating small and interpretable prediction sets may increase computational costs, especially for large-scale datasets or complex tasks.

---

### Official Review · Reviewer_8GnJ · 2025-03-01
**Interesting research, some details need to be described**

**Rating:** 6
**Confidence:** 3

**Review:**

This paper introduces a novel framework for conformal structured prediction, which extends traditional conformal prediction techniques to handle complex structured outputs such as code generation and question answering. The authors propose a method that constructs structured prediction sets represented as subsets of directed acyclic graphs (DAGs), enabling uncertainty quantification in settings where the label space cannot be simply expressed as a collection of regression or classification outputs. The approach is validated on two tasks: date-based question answering using SQuAD and Python code generation from the MBPP dataset.

Strengths:
- The proposed framework addresses an important gap in conformal prediction by extending it to structured prediction problems. This is particularly relevant for applications involving hierarchical labels or complex outputs like programs.
- The algorithm can be applied to various domains with structured outputs, including text and code generation, and provides both marginal and PAC (Probably Approximately Correct) coverage guarantees.
- The authors conduct experiments on two distinct tasks, demonstrating the effectiveness of their approach in constructing prediction sets that satisfy desired coverage guarantees while maintaining reasonable sizes.

Weaknesses:
- While the integer programming formulation for computing structured prediction sets is elegant, its computational cost is not thoroughly discussed. For large DAGs or high-dimensional label spaces, solving this optimization problem may become prohibitively expensive.
- Although the authors compare their approach to a baseline adapted from Khakhar et al. (2023), the discussion could benefit from comparisons with other state-of-the-art methods for uncertainty quantification in structured prediction tasks. Additionally, more details about the baseline implementation would help clarify its strengths and limitations relative to the proposed method.
- The sensitivity of the hyperparameters m, ϵ, and δ is analyzed qualitatively but lacks deeper exploration. A more comprehensive study of how these parameters interact and influence the trade-off between coverage and prediction set size would enhance the practical utility of the framework.

The paper makes a valuable contribution by proposing a flexible and theoretically grounded framework for conformal structured prediction. It successfully demonstrates the ability to construct interpretable prediction sets for complex outputs while satisfying coverage guarantees. However, addressing the computational complexity, expanding the scope of experiments, and providing a more thorough comparison with existing methods would further solidify the impact of this work. With these improvements, the paper has the potential to advance the field of uncertainty quantification for structured prediction tasks.

---

### Official Review · Reviewer_xcDJ · 2025-03-02
**Nice paper extending conformal prediction to structured label spaces.**

**Rating:** 6
**Confidence:** 4

**Review:**

The paper extends conformal prediction (CP) to structured label spaces, providing marginal coverage and PAC-style guarantees. The framework is described in sufficient detail along with a practical algorithm to estimate thresholds for conformal prediction. Empirical evaluation on the SQUaD dataset shows its practicality and effectiveness.

Extending CP to structured prediction settings is an important direction and it can be useful in LLMs, especially when dealing with structured outputs like code, reasoning traces, etc.  I have the following feedback to potentially improve the paper,

1. It is not clear how the structure of the space is taken into account. There are distances between the nodes in a DAG or any other structured space, where are they playing a role in CP?

2. The paper dives into technical details and loses touch with the application/example presented in Figure 1. It would help if the mathematical details and notations are also explained/introduced more clearly with examples.

3. Please clarify if you are outputting a set of graphs or a set of nodes from the graph and how do you get scores for each label?

4. Experiments on code generation can be helpful, especially when the motivating example (Figure 1.) is on code.

---

### Decision · Program_Chairs · 2025-03-04

Accept